# Control of *Fusarium* Head Blight of Wheat with *Bacillus velezensis* E2 and Potential Mechanisms of Action

**DOI:** 10.3390/jof10060390

**Published:** 2024-05-30

**Authors:** Jianing Ma, Chen Gao, Meiwei Lin, Zhenzhong Sun, Yuhao Zhao, Xin Li, Tianyuan Zhao, Xingang Xu, Weihong Sun

**Affiliations:** 1School of Agricultural Engineering, Jiangsu University, Zhenjiang 212013, China; majianing7581@163.com (J.M.); 2212216007@stmail.ujs.edu.cn (C.G.); 2212216002@stmail.ujs.edu.cn (M.L.); zhaoyuhao55@163.com (Y.Z.); 2212316010@stmail.ujs.edu.cn (X.L.); zhaotianyuan0711@163.com (T.Z.); 2212016006@stmail.ujs.edu.cn (X.X.); 2Jiangsu Suhe Socialized Agriculture Service Co., Ltd., Nanjing 210012, China; zhenzhong1250@163.com

**Keywords:** wheat, deoxynivalenol, *Fusarium asiaticum*, *Bacillus velezensis*, controlling of *Fusarium* head blight

## Abstract

Wheat plants are impacted by *Fusarium* head blight (FHB) infection, which poses a huge threat to wheat growth, development, storage and food safety. In this study, a fungal strain was isolated from diseased wheat plants and identified as *Fusarium asiaticum* F1, known to be a member of the *Fusarium graminearum* species complex, agents causally responsible for FHB. In order to control this disease, new alternatives need to be developed for the use of antagonistic bacteria. *Bacillus velezensis* E2 (*B. velezensis* E2), isolated from a previous investigation in our laboratory, showed a notable inhibitory effect on *F. asiaticum* F1 growth and deoxynivalenol (DON) synthesis in grains. The spore germination of *F. asiaticum* F1 was significantly reduced and the spores showed vesicular structures when treated with *B. velezensis* E2. Observations using scanning electron microscopy (SEM) showed that the hyphae of *F. asiaticum* F1 were shrunken and broken when treated with *B. velezensis* E2. The RNA-seq results of F1 hyphae treated with *B. velezensis* E2 showed that differentially expressed genes (DEGs), which were involved in multiple metabolic pathways such as toxin synthesis, autophagy process and glycan synthesis, especially the genes associated with DON synthesis, were significantly downregulated. In summary, those results showed that *B. velezensis* E2 could inhibit *F. asiaticum* F1 growth and reduce the gene expression of DON synthesis caused by F1. This study provides new insights and antagonistic mechanisms for the biological control of FHB during wheat growth, development and storage.

## 1. Introduction

*Fusarium* head blight (FHB) has caused great agricultural and economic losses worldwide. Wheat yields were reduced by 20 to 60 percent when severe FHB epidemics occurred in some regions [1]. In the USA, China, the UK, Africa, Brazil and elsewhere, severe FHB epidemics occur at a minimum of every fourth or fifth year. In the USA, yield losses as a result of FHB were estimated to be worth USD 3 billion between the early 1990s and 2008 [2]. FHB is mainly caused by the *Fusarium* graminearum complex (FSGC), which includes *F*. *asiaticum*, *F*. *graminearum*, *F*. *meridionale*, *F*. *ussurianum*, *F*. *boothii*, *F*. *nepalense*, *F*. *mesoamericaum*, *F*. *vorosii*, *F*. *louisianense*, *F*. *aethiopicum*, *F*. *brasilicum*, *F*. *gerlachii*, *F*. *acaciae-mearnsii*, *F*. *cortaderiae* and *F*. *austroamericanum* [3]. *Fusarium graminearum* is the main pathogen in FSGC. It may exist in all stages of grain growth and can spoil grains, rendering them inedible, by interfering with grain field growth and harvest storage.

*Fusarium* mycotoxins are a group of toxic secondary metabolisms secreted by *Fusarium* spp., such as DON, nivalenol and modified mycotoxins (3AcDON,15AcDON) [4,5,6], among which DON has the largest contamination range and amount in terms of grain. DON is synthesized during the infection of wheat by the *Fusarium* graminearum complex and is difficult to degrade under high temperatures and pressures. DON can pose a great threat to human and animal health, alter normal immune function and be serious enough to cause death [7]. In many countries, the detection rate of DON in wheat generally exceeds 50%. Therefore, DON on wheat has raised more attention regarding food safety incidents, leading many countries to introduce stringent foodstuff regulatory restrictions [6,7,8,9].

The conventional prevention and control methods for FHB include the chemical control and cross-breeding [10,11]. At present, chemical fungicides are most widely used to control FHB, but they are not eco-friendly and may cause resistance of the pathogen [10]. In contrast, biological control is eco-friendly and a cheap way to control these diseases [12]. Antagonistic *Bacillus* is one of the main research hotspots in the biological control of head blight [13]. At present, the *Bacillus* species widely used in biological control mainly include *Bacillus subtilis* [14], *Bacillus amyloliquefaciens* [15,16], *Bacillus velezensis* [17] and a combination of compatible biocontrol agents [18]. Antagonistic *Bacillus* is widely used in the biological control of other crops, such as maize [19], citrus [20], loquat [21], pepper and tomato [22]. For example, *Bacillus velezensis* S1 and S6 isolated from wheat ears showed bio-antimicrobial activity against the fungal pathogen of Septoria tritici blotch (*Zymoseptoria tritici*) [23]. The lipopeptide mycosubtilin from the beneficial bacterium *Bacillus subtilis* could also protect wheat against *Z. tritici* through a dual mode of action (direct and indirect), such as the priming of plant defense-related mechanisms [24]. Studies have found that the inhibition mechanisms of *Bacillus amyloliquefaciens* on *Fusarium oxysporum* include the inhibition of cell wall and membrane synthesis, the induction of increased membrane permeability and the destruction of ribosomes and mitochondria [25].

The purpose of this research was to study the antagonistic effect and mechanism of *B. velezensis* E2 on the pathogenic fungus of FHB and to find a biological approach to controlling FSGC. Therefore, this study was carried out with the following objectives: (1) To isolate and identify the main pathogen of FHB from the wheat region of the middle-lower Yangtze River; (2) To study the inhibitory effect of *B. velezensis* E2 on the pathogen by using scanning electron microscopy (SEM) (JEOL, Beijing, China); (3) To study the differentially expressed genes (DEGs) associated with *Fusarium asiaticum* F1 and their associated metabolic pathways under the influence of *B. velezensis* E2 using RNA sequencing and transcriptional analysis.

## 2. Materials and Methods

### 2.1. Wheat

The wheat variety Zhenmai 15 (*Triticum aestivum* L.) was sampled at Jiangsu Runguo Agricultural Development Co., Ltd., Zhenjiang City, Jiangsu Province, China. The wheat variety is susceptible to FHB, and the disease is easily contracted from the seedling stage to the panicle stage. Wheat plants and rhizosphere soil were multipoint-sampled during the flowering period in fields with severe FHB disease. The samples were used for the culture and isolation of the pathogen.

### 2.2. Isolation, Identification and Pathogenicity Verification of the Fungal Pathogen

The diseased wheat kernels that were characterized by a shriveled, slightly pinkish color were first picked out. The wheat kernels were soaked in 1% sodium hypochlorite solution for 3 min and then rinsed three times with sterile water. The diseased wheat kernels were briefly placed in a tube, and 10 mL of sterilized distilled water was added, followed by shaking in a shaker for 1 h. Gradient dilution was carried out in an ultra-clean workbench. Yang’s method [26] was applied, with slight modifications. Single colonies were isolated and purified by plate streaking on the potato dextrose agar (PDA) plate. DNA was extracted after multiple purification cultures. *F. asiaticum* F1 genomic DNA was extracted using the Shanghai Shenggong Fungal Genomic DNA Rapid Extraction kit (SK8229, Sangon Biotech, Shanghai, China). Fungi were identified by PCR amplification and sequencing with primers specific for the internal transcribed spacer (*ITS*) and transcription elongation factor (*TEF-α*) genes. The primer sequences are shown in Table 1. Afterwards, sequence alignments were performed according to the BLAST program of the National Center for Biotechnology Information (http://www.ncbi.nlm.nih.gov, accessed on 24 May 2023). 

F1 was cultured in carboxylmethyl cellulose medium, and the cultured conidia suspension was sufficiently diluted in sterile distilled water and counted using a hemocytometer. The final concentration was adjusted to 1 × 10^5^ spores/mL using a hemocytometer. A pathogenicity experiment caused by the isolated fungus was carried out according to the method of Wang [27]. By using the spikelet single-flower drip method, 10 μL 1 × 10^4^ spores/mL F1 spores was injected into each ear of wheat, with an equal amount of sterile water as a control. The samples were wrapped in plastic wrap and placed in an incubator, and the ears were removed after 7 days. Each treatment was repeated three times.

### 2.3. Analysis of the Toxigenic Chemotype of F1

With reference to Wang’s method [28], the specific primers were used to detect the toxigenic chemotype of *Fusarium*. The selected primers are shown in Table 2. The PCR reaction procedure of the primers is as follows: 95 °C, 4 min; 94 °C, 1 min; 58 °C, 50 s; 72 °C, 50 s; 30 cycles; 72 °C, 10 min. PCR products were added to 1% concentration gel sampling holes. After electrophoresis, the gel plate was placed in the ultraviolet transmissometer for detection.

### 2.4. Culturing the Potential Antagonist

The antagonistic bacteria obtained by screening in our laboratory was stored in the China Center for Type Culture Collection (CCTCC). The preservation number was CCTCC NO: M 2022579. The recommended classification name was *B. velezensis* E2. The antagonistic bacteria culture and concentration adjustment procedures were performed using Li’s method [29]. *B. velezensis* E2 was inoculated into LB medium at a 1% inoculation rate, cultured at 28 °C and 180 rpm for 18 h and then transferred to the new LB medium at the same inoculation rate. It was activated three times, and the final concentration was adjusted to 1 × 10^9^ cfu/mL.

### 2.5. Effect of B. velezensis E2 on F. asiaticum F1 Growth

The plate confrontation method was carried out on the potato dextrose agar (PDA) plate [30]. A hole punch with a diameter of 6 mm was used to drill four holes at equal spacing about 20 mm from the center of the PDA plate, and the agar in the holes was removed. Then, 30 μL 10^9^ cfu/mL *B. velezensis* E2 suspension was added to the left and right holes, and sterile water was added to the other two holes as the control. An *F. asiaticum* F1 plug with a diameter of 6 mm was placed in the center of the PDA plate. Then, the diameter of the pathogenic fungal hyphae was observed and determined after 7 d. There were three replicates, and the experiment was repeated twice.

The conidia of *F. asiaticum* F1 were obtained using the carboxylmethyl cellulose medium, and the concentration of spores was increased after brief centrifugation and resuspension. Then, 500 μL 1 × 10^9^ cfu/mL of the *B. velezensis* E2 suspension and 500 μL 1 × 10^6^ spores/mL of the F1 conidia suspension were added to 25 mL of potato dextrose broth (PDB), and an equal amount of sterile water was added as the control. After shaking at 180 rpm for 6.5 h, the spore germination rate and germ tube length of about 200 *F. asiaticum* F1 spores were statistically measured. If the length of the germ tube is at least half the length of the conidium, a conidium is regarded as having germinated. The spore germination rate was calculated according to the following formula.
(1) Spore germination rate (%) = (Number of germinated spores/200) × 100%


There were three replicates for each treatment, and the experiment was repeated twice.

### 2.6. SEM Analysis

Meng’s method was used, with slight modifications [31]. The collected *F. asiaticum* F1 hyphae were fixed in glutaraldehyde solution at 4 °C overnight. The specimens were washed three times with 0.1% moL/L phosphate-buffered saline for 15 min each time. Then, 30, 50, 70, 90 and 95% ethanol were used to dehydrate the specimens in gradients for 15 min each. The specimens were dehydrated with 100% ethanol three times for 20 min each time. Next, the ethanol was replaced with pure tertiary alcohol three times, and the standing time for each session was 15 min. Finally, the mixed mycelial pellet and tert-butanol suspension were sucked and dropped on the sample table covered with a cover glass. They were vacuum-dried in a freeze dryer that was pre-cooled for 1 h. The specimens were taken out after the air pressure dropped below 10 Pa. The dehydrated specimens were coated with gold-palladium and observed under the thermal field emission SEM (JSM-7001F).

### 2.7. Effect of Bacillus velezensis E2 on Toxin Accumulation in Wheat

A total of 50 g of wheat seeds was placed in a 150 mL conical flask. After high-temperature autoclave sterilization, 1 mL 1 × 10^5^ spores/mL F1 spore suspension was added to the conical flask. After 24 h, 3 mL 1 × 10^9^ cfu/mL *B. velezensis* E2 suspension was added to the conical flask, and an equal amount of sterile water was added as a control. There were three replicates for each treatment, and the experiment was repeated twice. The samples were kept at room temperature and in the dark for 30 days. Referring to the national food safety standard GB5009.111-2016 [32], the enzyme-linked immunosorbent assay (ELISA) screening method was used to detect the DON content in wheat grains. The method referred to the instructions of the vomiting toxin enzyme-linked immunosorbent assay kit (Qingdao Purebon Biotechnology Co., Ltd., Qingdao, China).

### 2.8. Sample Preparation and RNA Extraction

First, 100 µL 1 × 10^5^ spores/mL *F. asiaticum* F1 spore suspension was added to 50 mL PDB medium and cultured at 28 °C and 180 rpm for 18 h. The inoculum was inoculated into the new PDB medium according to the same inoculum amount and incubated for 18 h under the same conditions. *B. velezensis* E2 was activated in the LB medium and incubated at 28 °C and 180 rpm in a shaker protected from light for 18 h. After that, it was inoculated into the new LB medium according to the 1% inoculum and incubated for 18 h under the same conditions. The above two kinds of bacteria were centrifuged, and then the bacterial body was collected and dissolved in 100 mL PDB medium, while the control group was the same amount of *F. asiaticum* F1 body dissolved in 100 mL of the PDB medium.

All the above samples were repeated three times (CK1, CK2, CK3, T1, T2 and T3, respectively). After incubation in a shaker at 28 °C and 180 rpm for 10 h in the dark, the organisms were collected by centrifugation at 5000 rpm for 5 min and washed three times with PBS buffer, and then the samples were frozen in liquid nitrogen and stored at −80 °C at an ultra-low temperature in the refrigerator. The samples were pre-cooled with liquid nitrogen, and the total RNA was extracted according to the instructions of the RNA kit. The concentration, purity and integrity of RNA were detected by a microspectrophotometer.

### 2.9. Transcriptome Sequencing and DEGs Obtention for F1

The concentration, purity and integrity of the RNA in the sample were verified with Agilent 2100, and the qualified samples were sent to Gene Denovo Biotechnology Co. (Guangzhou, China) for the construction and sequencing of the cDNA libraries. After sequencing, a Clean Data sequence was assembled to obtain the unigene library. Then, the quality of the library was evaluated, and the gene expression quantity was carried out. DESeq2 software (v1.44.0) was used to screen differentially expressed genes. The genetic parameter of a false discovery rate (FDR) ≤ 0.05 and an absolute fold change (FC) ≥ 4 were considered to be differentially expressed genes.

### 2.10. GO and KEGG Classification

The expressed genes (FDR ≤ 0.05 and absolute FC ≥ 4) were functionally annotated according to three databases, the Kyoto Encyclopedia of Genes and Genomes (KEGG) and Gene Ontology (GO). Moreover, both upward and downward gene expression profiles were subjected to GO and KEGG enrichment analysis. In addition, GO and KEGG enrichment analyses were performed on both the upward and downward gene expression profiles.

### 2.11. Gene Expression Validation by RT-qPCR

#### 2.11.1. Primer Design

Partial differential genes were selected from the transcriptome analysis results for reference using the method of Hsu et al., and Primer design software Primer 5 (v5.5.0) was used to design primers based on gene sequence information [33]. The *cpc-1* gene and *rhoA* gene were used as an internal control. The specific information is shown in Table 3. The RNA obtained was used for reverse transcription. 

#### 2.11.2. RT-qPCR Reaction System and Procedure

Reverse transcription-quantitative polymerase chain reaction (RT-qPCR) was carried out to analyze the gene expression of *F. asiaticum* F1. In this section, a 20 μL reaction system was used, and the parameters in the real-time fluorescence quantitative PCR analyzer were set according to the method of Xu, as follows [34]: pre-denaturation for 90 s at 95 °C, denaturation for 5 s at 95 °C, annealing for 15 s at 60 °C and, finally, elongation for 20 s at 72 °C for 40 cycles; The dissolution curve was: 95 °C, 15 s; 60 °C, 1 min; 95 °C, 15 s. The procedures were performed according to the Vazyme Reverse Transcription Kit (R223) instructions. The RT-qPCR experiment was repeated three times. The relative expression level of the genes was calculated using the 2^−ΔΔCT^ method, and the standard deviation was calculated between three biological replicates.

### 2.12. Statistical Analyses

The experimental data were processed and analyzed using Excel 2021 and SPSS Statistics 18. A significant difference in the means was defined by Tukey’s test as *p* < 0.05.

## 3. Results

### 3.1. Isolation, Identification and Pathogenicity Verification of FHB Pathogens

In this section, the initial form of the pathogen is shown in Figure 1a. Because FSGC could not be accurately identified based on the *ITS* sequence, the phylogenetic tree was constructed based on the *TEF-1α* sequence. The acquired *TEF-α* rDNA gene sequence was analyzed using the Blast program on NCBI (http://www.ncbi.nlm.nih.gov, accessed on 24 May 2023). There was more than 99% homology between the F1 strain and *F. asiaticum* strains in the NCBI database. The phylogenetic tree was created, as depicted in Figure 1b. As shown in Figure 1c, the wheat seeds turned red after a month of F1 infection, and the wheat was fully rotten after 2 weeks of F1 infection. Based on the morphological characteristics and molecular biological identification, the pathogen F1 was identified as *F. asiaticum*.

### 3.2. Analysis of the Toxigenic Chemotype of F1

The DNA of F1 was amplified by using primers for detecting DON and NIV. As shown in Figure 2, the amplified fragment of F1 was about 200 bp, indicating that F1 produced the DON toxin. Subsequently, the primers for detecting 3-AcDON and 15-AcDON were used to amplify F1, and the amplified fragment of F1 was about 586 bp, indicating that F1 produces 3-AcDON. There were no bands at 864 bp, indicating that F1 did not produce 15-AcDON. This indicated that the toxigenic chemotypes of F1 were DON and 3-AcDON.

### 3.3. Antagonistic Activity of Bacillus velezensis

As shown in Figure 3a, the radial growth of *F. asiaticum* F1 on a PDA plate was inhibited by *B. velezensis* E2. As shown in Table 4, the spore germination rate of F1 treated with *B. velezensis* E2 was only 25.17%, while that of the control group was 93.33%, indicating that *B. velezensis* E2 could effectively inhibit the spore germination of *F. asiaticum* F1. Figure 3b showed that a vesicle-like structure was formed at the end of the spore germ tube with the treatment of *B. velezensis* E2. These results demonstrated the inhibitory effect of *B. velezensis* E2 on *F. asiaticum* F1.

### 3.4. SEM

The results of the electron microscopy further showed the inhibitory effect of *B. velezensis* E2 on *F. asiaticum* F1. As shown in Figure 4, under the influence of *B. velezensis* E2, the hyphae of *F. asiaticum* F1 exhibit shrinkage, twisting and breaking. The same changes also occurred in the mycelia morphology when *F. graminearum* was treated with myriocin [35].

### 3.5. Effect of B. velezensis E2 on DON Accumulation in Storage Wheat 

As shown in Table 5, compared with wheat treated with only F1 spore suspension (CK), the amount of DON in wheat treated with *B. velezensis* E2 (T) was significantly reduced after 30 days of storage. This indicates that *B. velezensis* E2 can inhibit the growth of F1, thereby reducing the generation of vomiting toxins on wheat.

### 3.6. Sequencing Data and Its Quality Control

As shown in Table 6, 40,583,875 high-quality reads were obtained in the treatment group (*F. asiaticum* co-cultivation with *B. velezensis* E2) and control group (*F. asiaticum* co-cultivation with sterile water) using Illumina HiSeq2500 (Illumina, San Diego, CA, USA). After filtering low-quality reads, the GC percentage was approximately 52.70%. Moreover, among the sequenced samples, the Q20 varied from 98.64% to 98.76%. In total, 87.48% of readings could be mapped uniquely with the reference genome.

### 3.7. Differential Expression Gene Analysis

The differentially expressed genes between the control group and the treatment group are shown in Figure 5. A total of 3482 differentially expressed genes were screened using |log_2_ (Fold Change)| ≥ 4, FDR ≤ 0.05 as the criteria, among which 2071 genes were down-regulated and 1411 genes were up-regulated.

### 3.8. GO Enrichment Analysis of Differentially Expressed Genes 

To examine the assignment and distribution of GO terms among the DEGs, GO enrichment analysis was carried out, as shown in Figure 6. There were 3414 genes which were annotated to the GO database among the DEGs. The GO functionality included biological processes, molecular function and cell components. These three categories contain 18, 10 and 11 GO terms, respectively.

There were 2817 DEGs in the biological process category. The category with the highest number of DEGs is the cellular metabolic process (769 DEGs). The other GO terms included the cellular process, single-organism process, multicellular organismal process, detoxification, growth, multi-organism process, reproduction, cellular component organization or biogenesis, reproductive process, localization, positive regulation of the biological process, signaling, developmental process, negative regulation of the biological process, response to the stimulus, biological regulation and regulation of the biological process.

In the molecular function category, 10 GO terms were enriched and included with 1181 DEGs. Most genes were enriched into catalytic activity, which included 581 DEGs. The other GO terms included the structural molecule activity, antioxidant activity, transporter activity, nucleic acid binding transcription factor activity, signal transducer activity, molecular function regulator, transcription factor activity, protein binding, molecular transducer activity and binding.

In the cell components category, 11 GO terms were enriched and included with 2144 DEGs. Most genes were enriched into the cell and cell part, and they have an equal number of DEGs. The other GO terms included the nucleoid, membrane part, membrane, membrane-enclosed lumen, supramolecular fiber, macromolecular complex, extracellular region, organelle and organelle part.

### 3.9. KEGG Enrichment Analysis of Differentially Expressed Genes

The KEGG enrichment and classification of differential genes showed that a total of 679 differential genes were involved in 117 metabolic pathways. Figure 7 shows that differential genes are mainly involved in metabolism, genetic information processing, environmental information processing, cellular processes and organismal systems. These primary metabolic pathways are divided into 21 secondary metabolic pathways, among which genomics processes include replication and repair, translation, transcription, etc. Cellular processes include transport and catabolism. Environmental information processes include membrane transport and signal transduction. The basal metabolic process includes lipid metabolism, the biosynthesis of other secondary metabolites and the metabolism of other amino acids. Biological systems involve environmental adaptation. The top 10 KEGG metabolic pathways with the most significant enrichment include the ribosome, biosynthesis of amino acids, metabolic pathways, biosynthesis of antibiotics, ribosome biogenesis in eukaryotes, gluconeogenesis and pyruvate metabolism, biosynthesis of secondary metabolites, N-Glycan biosynthesis and fructose and mannose metabolism.

#### 3.9.1. Effects on DON Synthesis Pathways

*F. asiaticum* F1 treated with *B. velezensis* E2 produced many different genes in the DON synthesis pathway, as shown in Table 7. The *Tri101*, *Tri10* and *Tri1* expression levels were down-regulated in the *F. asiaticum* F1 treated with *B. velezensis* E2. An analysis of the transcriptome data indicated that the gene *cut6*, which is involved in acetyl-CoA synthesis, was significantly down-regulated under the action of antagonizing *B. velezensis* E2.

#### 3.9.2. Effects on Secondary Metabolic Pathways

The genes related to mannose synthesis, such as *ALG9*, *alg-11*, *ALG12*, *dpm1* and *gls2*, were all down-regulated, as shown in Table 5. The gene *TCB1*, which is involved in the synthesis of transmembrane proteins, was also down-regulated, indicating that the *B. velezensis* E2 affected the cell membrane formation and material transport of the pathogenic fungi. Transcriptome data also showed that under the influence of *B. velezensis* E2, the expression of *ATG8* was up-regulated, and the expression of other genes related to autophagy protein synthesis such as *ATG2*, *ATG9*, *ATG12* and *ATG13* were also up-regulated. In addition, most of the DEGs related to ribosome biogenesis in eukaryotes pathways were up-regulated under *B. velezensis* E2 treatment. By analyzing the transcriptome data, it was found that the DEGs of *F. asiaticum* F1 related to ribosome synthesis were up-regulated, among which *mpp10*, *UTP14*, *UTP15* and *UTP4* were related to the synthesis of nucleolar RNA-related proteins, and *NOG1* was related to the synthesis of nucleolar GTP-binding proteins, as shown in Table 7. 

### 3.10. RT-qPCR Verification of Differentially Expressed Genes

RT-qPCR was used to verify the expression levels of the DEGs, as shown in Figure 8. The expression levels of *KRE33*, *ATG8*, *TRI101* and *ATG13* were up-regulated, while the expressions of *alg-11*, *ade5*, *TCB1* and *cut6* were down-regulated, which was basically consistent with the results of the transcriptional analysis. 

## 4. Discussion

As mentioned above, FHB causes serious losses in the wheat yield and quality worldwide every year. In China, the middle-lower reaches of the Yangtze River are the main regions of FHB. The Zhenjiang area belongs to the lower reaches of the Yangtze River. In this study, the sampling and screening of wheat in the Zhenjiang area were carried out, and the pathogen of wheat FHB was identified as *Fusarium asiaticum* by morphological characteristics and DNA sequencing. After analyzing the toxigenic chemotype of F1, it was found that its toxigenic chemotypes are DON and 3-AcDON. DON mainly contaminates wheat, seriously affecting the yield and quality of grain, feed and food and even causing serious food safety incidents. Studies have further shown that DON was the most abundant mycotoxin found in infected wheat heads, and the concentrations were consistently higher than those of its acetylated derivatives, 15-AcDON and 3-AcDON [27]. Therefore, it is important to find a suitable method for inhibiting the infection of *Fusarium asiaticum* and the accumulation of DON.

Many studies have found that *Bacillus* has an obvious inhibitory effect on *Fusarium*, so it is an important resource for the development of microbial fungicides [36,37]. However, there are few studies on the effect of *Bacillus* on toxin accumulation and the mechanism of inhibiting toxin production by *Fusarium*. In this study, *B. velezensis* E2, previously screened by our laboratory, was used to conduct related research. The plate confrontation experiment revealed that *B. velezensis* E2 had a significant inhibitory effect on *F. asiaticum* F1, and the inhibition effect of *B. velezensis* E2 on *F. asiaticum* F1 was further revealed through the spore germination experiment and scanning electron microscope observation. It was also found that *B. velezensis* E2 can significantly reduce the content of DON in wheat. Other studies have shown that *Bacillus amyloliquefaciens* degraded DON mainly by secreting extracellular enzymes [38]. In order to better understand the antagonistic mechanism of *B. velezensis* E2 against *F. asiaticum* F1, the hyphae of *F. asiaticum* F1 under *B. velezensis* E2 were studied by transcriptomics techniques.

The RNA-seq result showed that from the total gene reads, 3482 genes were differentially expressed (either up-regulated or down-regulated). Then, the DEGs were analyzed to determine their GO enrichment. The GO enrichment was highly concentrated in the metabolic process (769 DEGs), cellular processes (620 DEGs), single-organism process (614 DEGs) and catalytic activity (581 DEGs). Then, KEGG pathway enrichment analysis was conducted to determine the DEGs that have functional networks and biological pathways in the system. In the KEGG enrichment classification of DEGs, the DEGs in the treated F1 hyphae related to toxin synthesis, glycan synthesis, ribosome synthesis and the autophagy process were analyzed.

The FHB not only leads to a huge decrease in the wheat yield but also increases the risk of excessive levels of DON in the storage period of wheat [39]. Mycotoxins are secondary metabolites of fungi that cause serious damage to agricultural products and foods in the food supply chain. These harmful pollutants have been directly linked with poor socio-economic patterns and human health issues [40]. In the DON synthesis pathways, *B. velezensis* E2 treatment produced many different genes in the DON synthesis pathway, as shown in Table 5. The biosynthesis of DON is a series of complex processes, which are closely associated with the expression of a series of TRI genes [41]. The 12-gene core TRI cluster, the two-gene *TRI1-TRI16* locus and the single-gene *TRI101* locus all generate trichothecene biosynthesis enzymes [41,42]. *Tri1* plays a crucial role in DON synthesis. It encodes the cytochrome *P450* monooxygenase. *Tri6* and *Tri10* regulate the transcription of TRI genes [42]. An acetyltransferase that catalyzes the esterification of acetyl to the C-3 hydroxyl of trichothecenes is encoded by the gene *TRI101* [42]. Previous research by Xu showed that succinate dehydrogenase inhibitors (SDHIs) could decrease the DON biosynthesis of *F. asiaticum* [10]. This fungicide can inhibit the synthesis of DON, and its mechanism of action is mainly to inhibit the expression of key genes in glycolysis, thereby reducing the content of pyruvate and acetyl-CoA, the raw materials for DON synthesis. It can be seen in Table 7 that *Tri101*, *Tri10* and *Tri1*, related to DON synthesis, and *cut6*, related to acetyl-CoA synthesis, were all down-regulated in *F. asiaticum* F1 treated with *B. velezensis* E2. This suggested that *B. velezensis* E2 might inhibit DON synthesis by down-regulating the expression of genes related to enzymes and raw materials for DON toxin synthesis.

N-glycans or asparagine-linked glycans are the main components of eukaryotic glycoproteins, which are located on cell membranes. By analyzing the transcriptome data, it was found that the genes related to mannose synthesis, such as *ALG9*, *alg-11*, *ALG12*, *dpm1* and *gls2*, were all down-regulated, as shown in Table 5. The gene *TCB1*, which is involved in the synthesis of transmembrane proteins, was also down-regulated, indicating that the *B. velezensis* E2 affected the cell membrane formation and material transport of the pathogenic bacteria. This also explains the vesicle-like structure of the mycelium of *F. asiaticum* F1, observed under the microscope earlier, in the presence of *B. velezensis* E2. Peptidoglycan is an important part of the cell wall. It was found that under the treatment of *B. velezensis* E2, genes related to glycan synthesis in *F. asiaticum* F1 were down-regulated, suggesting that *B. velezensis* E2 could affect the formation of the cell wall. Many biocontrol microorganisms can inhibit the growth and reproduction of pathogenic fungi by producing glucanase and chitinase to destroy the integrity of the cell wall of pathogenic fungi [43]. Some studies have also shown that lipopeptide compounds produced by *B. velezensis* E2 can destroy the cell membrane and cell wall of pathogenic bacteria, which is consistent with the expression of several key genes above. Therefore, it was speculated that *B. velezensis* E2 destroyed the integrity of the cell wall and interfered with the composition of the membrane lipids of *F. asiaticum* F1 by affecting the expression of related genes involved in N-glycan synthesis. It was also an important reason why *B. velezensis* E2 inhibits the growth of *F. asiaticum* F1 mycelium. 

Ribosomes are large macromolecular assemblies, with approximately two-thirds of their mass consisting of RNA, with the rest being proteins [44], and their main function is to synthesize proteins. By analyzing transcriptome data, it was found that genes related to ribosome synthesis were up-regulated in *F. asiaticum* F1. Among those up-regulated genes, *mpp10*, *UTP14*, *UTP15* and *UTP4* were related to nucleolar RNA-related protein synthesis, and *NOG1* was related to nucleolar GTP-binding protein synthesis. This indicated that *F. asiaticum* F1 maintained protein synthesis by synthesizing ribosomes to resist the harmful effects of *B. velezensis* E2. Ma et al. also found that, in response to the stress of *Bacillus amyloliquefaciens*, the genes involved in the pathway of ribosome synthesis were significantly up-regulated in *Fusarium oxysporum* to alleviate the degradation of the protein by bacteriostatics [25]. The down-regulated gene *ade5* encodes a phosphoribosamine-glycine ligase. Kim et al. constructed a deletion mutant of *ade5* and found that its hyphal growth rate was slowed down and almost no conidia were produced [45]. This also explained why *B. velezensis* E2 could significantly inhibit the spore germination rate of *F. asiaticum* F1 from the molecular level.

Autophagy is a degradation pathway that removes cytoplasmic material from eukaryotic cells and is characterized by the formation of two-membrane structures called autophagosomes. The autophagy balance of a pathogen is very important for their growth, pathogenicity and environmental competitiveness [46]. Biocontrol *Streptomyces* sp. S89 inhibits the growth of scab hyphae by secreting the active substance rapamycin, reducing the pathogenicity of the pathogen and the synthesis of mycotoxins [47]. The effect of rapamycin on the Target of Rapamycin signaling pathway of *Gibberella* spp. could promote the proteasome-dependent degradation of histone acetyltransferase *Gcn5*, thereby reducing the level of acetylation of *Atg8* and promoting autophagy [47]. The expression of genes related to eukaryotic autophagy in the autophagy pathway of *F. asiaticum* F1 was up-regulated under the treatment of *B. velezensis* E2. It can be seen in Table 7 that *ATG2*, *ATG9*, *ATG12* and *ATG13*, involved in the autophagy protein synthesis, were all up-regulated. It is speculated that *F. asiaticum* F1 can defend against the damage of *B. velezensis* E2 by autophagy. 

## 5. Conclusions

In this study, the pathogen of FHB was identified as *F. asiaticum* F1, which can produce a toxic secondary metabolite named DON. It was found that *B. velezensis* E2 can not only inhibit the pathogen growth but also inhibit the accumulation of DON. Compared to the control group, a total of 3482 differentially expressed genes were screened by the transcriptome sequencing of *F. asiaticum* F1 from the *B. velezensis* E2 treatment group. Then, GO functional annotation and KEGG enrichment analysis were performed for these differentially expressed genes. The results showed that these differentially expressed genes were involved in DON synthesis, glycan synthesis, ribosome synthesis and the autophagy process. In addition, the RT-qPCR validation results of the differentially expressed genes were consistent with the basic trend of the sequencing results. These results provide a theoretical basis for further revealing the molecular mechanism of disease resistance induced by *B. velezensis* E2.

## Figures and Tables

**Figure 1 jof-10-00390-f001:**
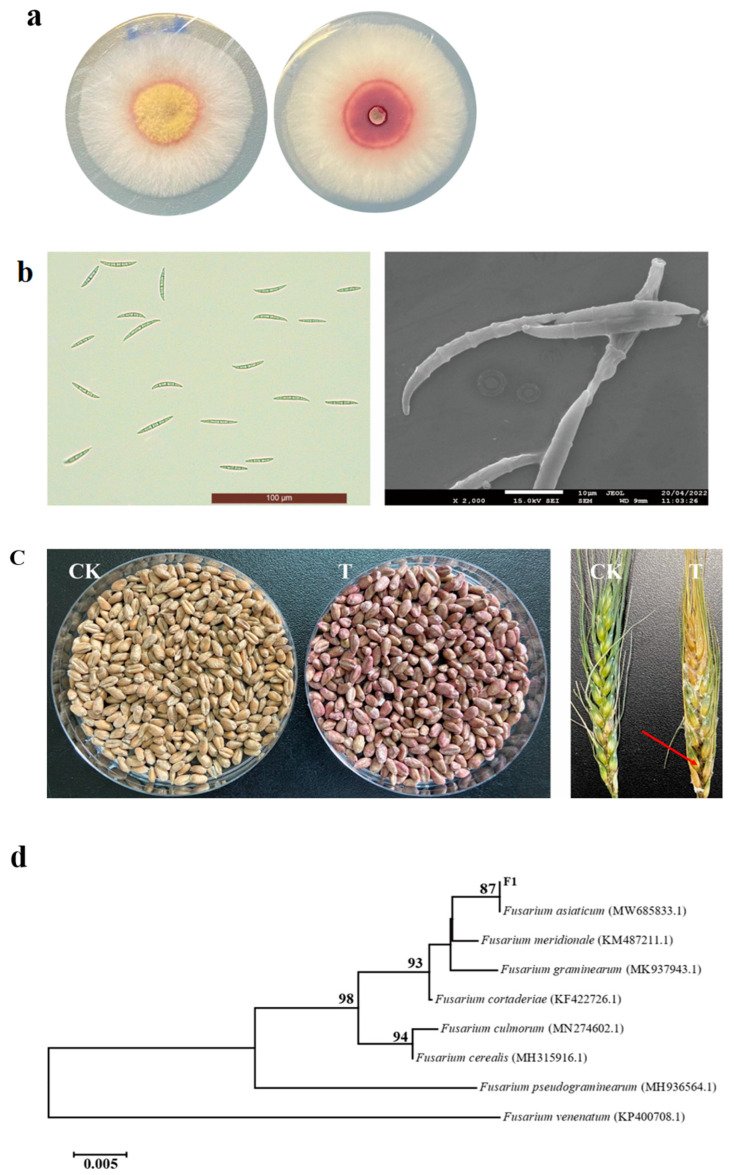
(**a**) The colony morphology of *F. asiaticum* F1 cultured on the PDA plate for 5 d; (**b**) Morphology of *F. asiaticum* F1 conidia under an optical microscope (left) and scanning electron microscope (right); (**c**) Verification of the pathogenicity of *F. asiaticum* F1 in storage (left) and growing wheat (right). Changes in wheat inoculated with the F1 spore suspension for one month in the storage period and 7 d in the growth period. The wheat seeds turned red (left); The spikes of wheat turned yellow and moldy. The area indicated by the arrow showed symptom of blackening (right). (**d**) Phylogenetic tree of the strain F1 based on the sequences of the *TEF-α* rDNA gene. Using MEGA 7.0 software, the phylogenetic tree was created using the neighbor-joining method. The branch points display the bootstrap values.

**Figure 2 jof-10-00390-f002:**
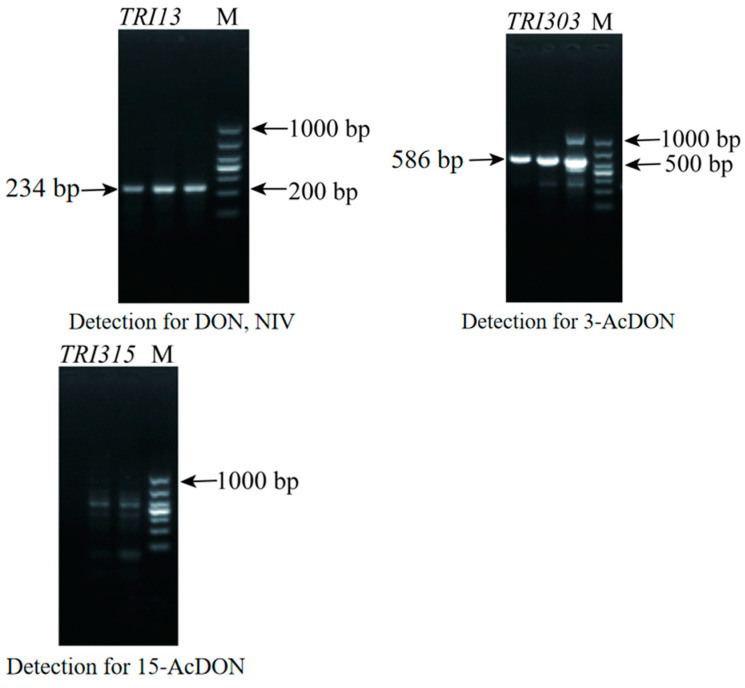
Determination of the F1 toxigenic chemotype by agarose gel electrophoresis.

**Figure 3 jof-10-00390-f003:**
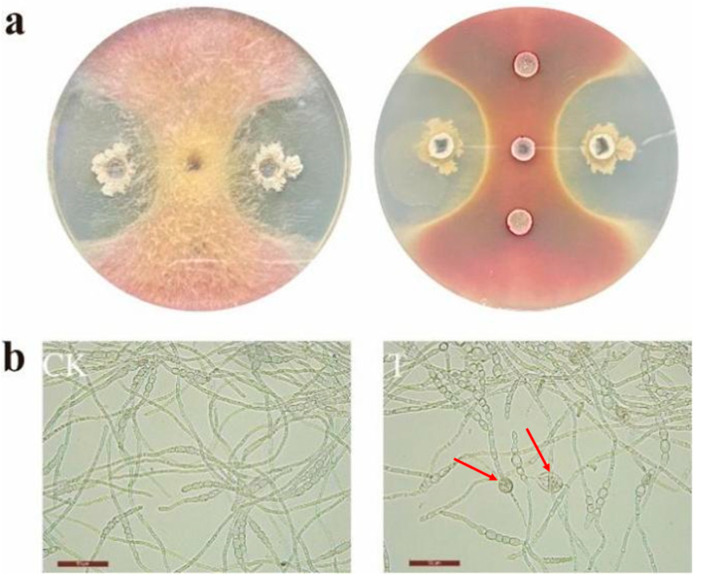
Antagonistic *B. velezensis* E2 on *F. asiaticum* F1. (**a**) Effect on the growth of *F. asiaticum* F1 after 7 d on the PDA plate. *B. velezensis* E2 suspension (30 μL) was added to the left and right holes, and an equal amount of sterile water was added to the upper and lower holes as a control. (**b**) Effect on the germination of *F. asiaticum* F1 spores. The area indicated by the arrow showed vesicle-like structures.

**Figure 4 jof-10-00390-f004:**
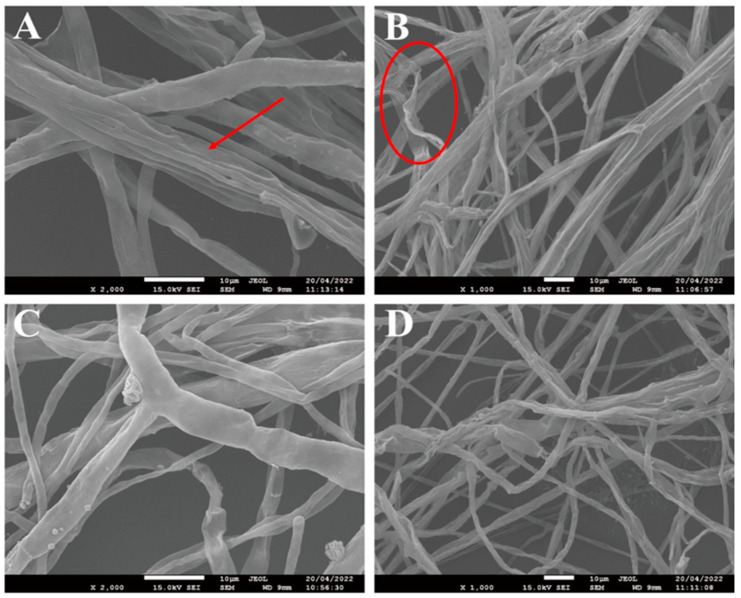
Scanning electron microscopy of *F. asiaticum* growth without the presence of *B. velezensis* E2 (control: (**C**,**D**)) and faced with *B. velezensis* E2 (**A**,**B**). The areas indicated by arrow and circle showed shrinkage and twisting of the F1 hyphae with the treatment of *B. velezensis* E2.

**Figure 5 jof-10-00390-f005:**
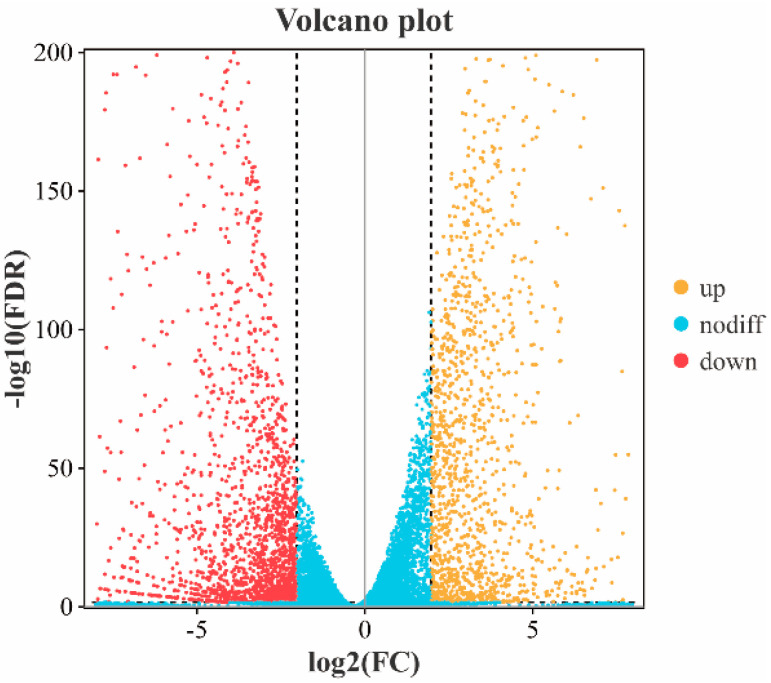
Volcano plot of differentially expressed genes. The horizontal coordinate of the volcanic map shows the difference multiple between the two groups, and the vertical coordinate shows the −log10 value of the FDR value of the difference between the two groups. Each dot in the volcanic map represents a gene, with the red representing up-regulated genes, the yellow representing down-regulated genes and the blue representing non-differentially expressed genes.

**Figure 6 jof-10-00390-f006:**
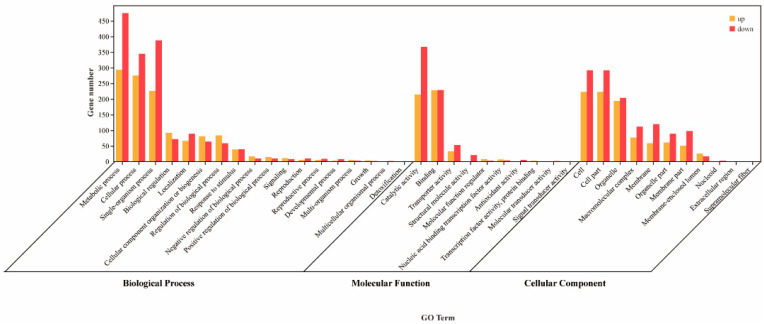
Classification of differentially expressed genes based on GO annotation. The horizontal coordinate of the bar chart shows the second-order GO term, and the vertical coordinate shows the number of differentially expressed genes in this term, with yellow indicating up-regulation and red indicating down-regulation.

**Figure 7 jof-10-00390-f007:**
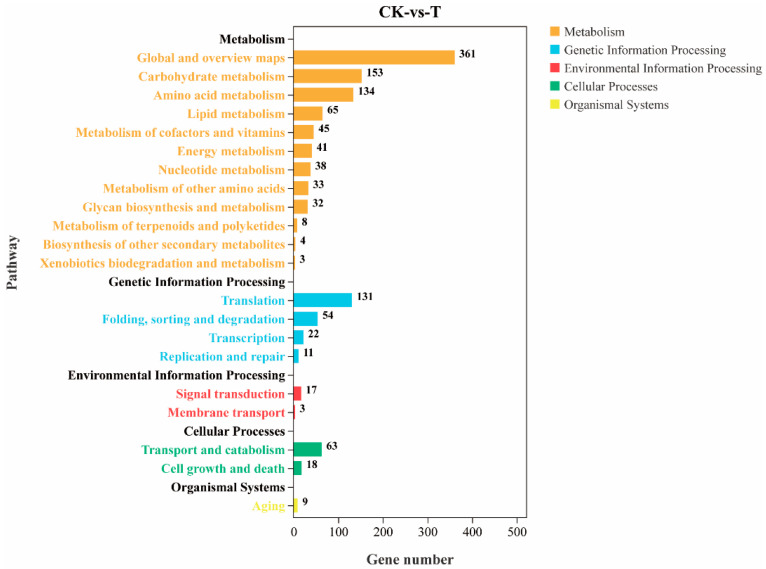
KEGG pathway classification of differentially expressed genes. The vertical coordinate represents primary and secondary metabolic pathways, and the horizontal coordinate represents the number of differentially expressed genes in this metabolic pathway.

**Figure 8 jof-10-00390-f008:**
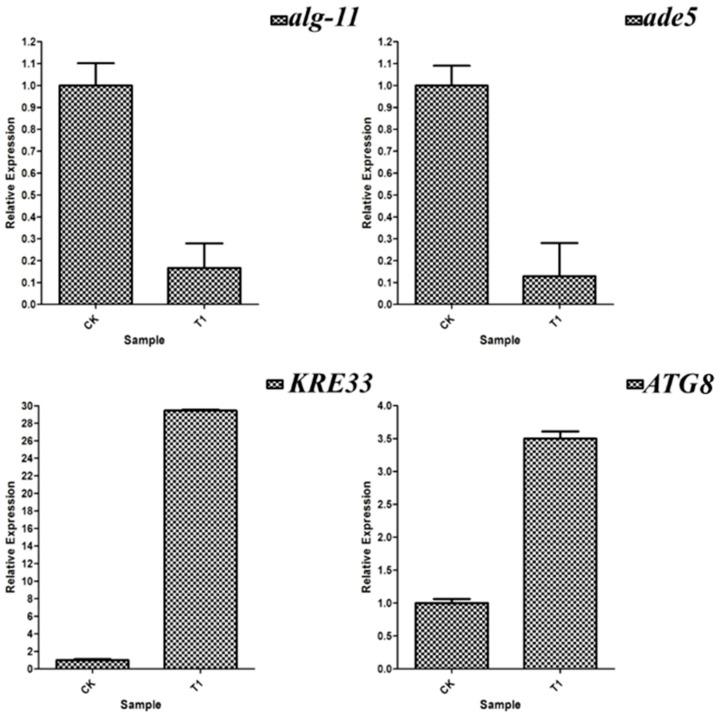
Results of the verification of some differentially expressed genes by RT-qPCR. The internal reference gene was *cpc-1* gene. The RT-qPCR experiment was repeated three times, and the error bar indicated the degree of dispersion.

**Table 1 jof-10-00390-t001:** Primer sequences.

Gene Name	Accession Number	Primer	Primer Sequence (5′→3′)	Product Length
*ITS*	NR_121320.1	ITS1	TCCGTAGGTGAACCTGCGG	559
		ITS4	TCCTCCGCTTATTGATATGC	
*TEF-α*	NW_022157945.1	TEF-F	ATGGGTAAGGAGGAGAAGAC	700
		TEF-R	GGAAGTACCAGTGATCATGTT	

Note: Accession number was from the NCBI database.

**Table 2 jof-10-00390-t002:** Primer sequences of the toxigenic chemotype.

Gene Name	Accession Number	Primer	Primer Sequence (5′→3′)	Toxigenic Chemotype	Product Length
*Tri13*	NC_058400.1	Tri13-F	TACGTGAAACATTGTTGGC	deoxynivalenol (DON), nivalenol (NIV)	234, 415
		Tri13-R	GGTGTCCCAGGATCTGCG		
*Tri3*	NC_058400.1	Tri303-F	GATGGCCGCAAGTGGA	3-AcDON (3-acetyldeoxynivalenol)	586
		Tri303-R	GCCGGACTGCCCTATTG		
		Tri315-F	CTCGCTGAAGTTGGACGTAA	15-AcDON (15-acetyldeoxynivalenol)	864
		Tri315-R	GTCTATGCTCTCAACGGACAAC		

Note: The accession number was from the NCBI database.

**Table 3 jof-10-00390-t003:** Primer design of differentially expressed genes.

Gene ID	Gene Name	Accession Number	Revers Primer (5′ to 3′)
FGSG_00732	*KRE33*	NC_026474.1	F: CGCAAAGCGGTGGACT R: GAATCTTGTCGGTCTCCTTGT
FGSG_06580	*cut6*	NC_026477.1	F: AGCGAGCCATTCACTTCACT R: TCCAGGGGGTCCGATAAAGA
FGSG_07896	*TRI101*	NC_026477.1	F: GTTCTGCCGTGCTGTTGATG R: GTCTCACAGTCTCGGGCTTAC
FGSG_08429	*ade5*	NC_026475.1	F: GGGCAGACAAAGCAGGCA R: TAGGCTCGCTCAATGGCAC
FGSG_08491	*ATG13*	NC_026475.1	F: AACGACCCAGCCGAATCTAC R: ACTGCTCCATCACATCCCAC
FGSG_10740	*ATG8*	NC_026476.1	F: GAGGTTCTACCCCCGACAG R: CGCCAAAAGTGTTCTCGCC
FGSG_10858	*ALG-11*	NC_026476.1	F: CCGACCCGAGAAGAACCATC R: CTCAGCCAGTCCAGAACCTC
FGSG_06885	*TCB1*	NC_026477.1	F: TCAAGGGCGAGGATGGAC R: GGCAGGTCGGAACAGAAGTC
FGSG_09286	*cpc* *-1*	NC_026477.1	F: GCCTTTTCCTCACCTGCTGT R: CCGACTTGCGACGGTTCA
FGSG_04400	*rhoA*	NC_026475.1	F: GGCGATGGTGCTTGTGGTAA R: GAGGGAGTCGGGAGAGTCAA

Note: Accession number was from the NCBI database.

**Table 4 jof-10-00390-t004:** Effect of *B. velezensis* E2 on the germination of *F. asiaticum* F1 spores.

Treatment	CK	*B. velezensis* E2
Spore germination rate	93.33 ± 0.76	25.17 ± 3.32 **

The results indicated the mean ± standard deviation and were tested by an independent sample T test, ** *p* < 0.01.

**Table 5 jof-10-00390-t005:** Effect of *B. velezensis* E2 on the deoxynivalenol production of *F. asiaticum* F1.

Experiment	Control Group	Treatment Group
DON content (μg/kg)	46.02 ± 1.19	36.72 ± 0.83 **

The results indicated the mean ± standard deviation and were tested by an independent sample T test, ** *p* < 0.01.

**Table 6 jof-10-00390-t006:** Sequencing data tables of *F. asiaticum* F1.

Samples	Raw Data (bp)	Clean Data (bp)	Clean Reads	GC (%)	Q20 (%)	Unique-Mapped (%)
CK1	6,112,122,000	6,075,824,999	40,647,974	52.91	98.70	35,427,322 (87.32%)
CK2	6,529,115,400	6,479,907,696	43,433,248	52.96	98.69	38,186,463 (88.06%)
CK3	6,884,337,000	6,836,297,207	45,781,106	53.10	98.65	40,361,865 (88.30%)
T1	5,622,680,700	5,585,596,453	37,401,952	52.36	98.72	32,604,351 (87.36%)
T2	5,586,865,500	5,542,412,171	37,159,506	52.26	98.76	32,255,302 (86.95%)
T3	5,876,491,800	5,830,758,067	39,079,466	52.58	98.64	33,887,979 (86.88%)

**Table 7 jof-10-00390-t007:** Effects of *B. velezensis* E2 on *F. asiaticum* F1 genes related to DON synthesis.

Gene ID	Gene Name	log_2_FC	Definition
FGSG_07896	*TRI101*	−5.02	richothecene 3-O-acetyltransferase
FGSG_03538	*TRI10*	−10.75	TRI10 [Fusarium graminearum]
FGSG_00071	*TRI1*	−6.28	cytochrome P450 monooxygenase
FGSG_06580	*cut6*	−4.3	acetyl-CoA carboxylase
FGSG_05243	*ALG9*	−3.13	alpha-1,2-mannosyltransferase
FGSG_10858	*alg-11*	−2.24	alpha-1,2-mannosyltransferase
FGSG_01233	*ALG12*	−3.25	alpha-1,6-mannosyltransferase
FGSG_04044	*dpm1*	−2.37	dolichol-phosphate mannosyltransferase
FGSG_05973	*gls2*	−3.28	alpha-glucosidase
FGSG_06885	*TCB1*	−2.62	transmembrane protein
FGSG_08429	*ade5*	−2.64	phosphoribosylglycinamide formyltransferase
FGSG_10740	*ATG8*	4.44	GABA(A) receptor-associated protein
FGSG_08491	*ATG13*	3.34	autophagy-related protein 13
FGSG_10033	*mpp10*	3.56	U3 small nucleolar RNA-associated protein
FGSG_08759	*UTP14*	4.90	U3 small nucleolar RNA-associated protein 14
FGSG_10060	*UTP15*	3.70	U3 small nucleolar RNA-associated protein 15
FGSG_00270	*UTP4*	4.69	U3 small nucleolar RNA-associated protein 4
FGSG_06165	*NOG1*	3.57	nucleolar GTP-binding protein

## Data Availability

The raw sequence data reported in this paper have been deposited in the Genome Sequence Archive (Genomics, Proteomics & Bioinformatics 2021) in the National Genomics Data Center (Nucleic Acids Res 2022), China National Center for Bioinformation/Beijing Institute of Genomics, Chinese Academy of Sciences (GSA: CRA013405), and they are publicly accessible at https://ngdc.cncb.ac.cn/gsa (accessed on 12 November 2023).

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
