# Peer review of "Control of Fusarium Head Blight of Wheat with Bacillus velezensis E2 and Potential Mechanisms of Action"

_jof, 2024, doi:10.3390/jof10060390_

Round 1

Reviewer 1 Report (Previous Reviewer 1)

The authors responded appropriately to the reviews. I think that the paper could be published.

None

Author Response

Reviewer 2 Report (Previous Reviewer 2)

 The aim of this project is to isolate and identify a bacterial agent capable of controlling the development of Fusarium asiaticum, known to be part of the fungal complex responsible for FHB (Fusarium head bllight). A pathovar of the fungus was recently isolated from wheat fields and its sound characterization of the toxigenic chemotype was carried out. In order to test the bacterium's ability to limit the development of the disease on wheat, the authors tested the bacterium's ability to inhibit the growth of the fungus, as well as its ability to produce toxins. 

This work is interesting in view of the importance of developing alternative solutions against impacting wheat diseases, such as FHB. I recommend the publication of this article with a few corrections.

Here are a few comments that will help us to further improve this important work.

Introduction Part :

Paragraph L57-L61 : thanks to check if B. velezensis or B. amyloliquefaciens and their product have not been already used to control another disease on wheat, thanks to add informations about that, for example two articles, but probably ….. thanks to check if other disease on wheat have been not already …

If I‘ll remember, it’s the case against the STB : Septoria Triciti Blotch, caused by the hemibiotrophic fungi : Zymoseptoria tritici. For example, B. velezensis against STB : Platel et al 2022 Isolation and identification of lipopeptide-producing Bacillus velezensis strains from wheat phyllosphere with antifungal activity against the wheat pathogen Zymoseptoria tritici. Agronomy. 12: 95. And Lipopeptides products by B. species (B. amyloliquefaciens) :  Platel et al.2023 : Deciphering immune responses primed by a bacterial lipopeptide in wheat towards Zymoseptoria tritici. Front. Plant Sci. 13:1074447

Mat et Meth part :

L 75 : 2.1 Wheat : Please add a stage of sensitivity of the cultivar used to test the pathogenicity of the strain (highly sensitive, sensitive, not very sensitive cultivar.... etc)

Table 1, Table 2 and Table 3 : Please add the references of the genes targeted by the primers used : NCBI referensce or other database

Results :

Please complete the figure legends with the stats genes used and add the asterisk or other sign showing statistically different data (Fig 8)

Discussion Part  :

L385-386 : Wheat yields …. occur” must be place in the introduction part

L386 : The sentence : “the disease is mainly caused by FSGC” have been to be eliminate.

L386-387 : the link between the two sentences : the Zhenjiang area is a area close to the Yagtze River ? if not, for me you can transfer the sentence “In China….. FHB.” In the introduction part. If it’s the same area, thanks to leave it here.

Minor revision :

L44 : “nivalenol” and not “Nivalenol”

L 53 : add a “and” :  “the chemical control and molecular breeding”

L53 : have you sure of your reference number 9 to review the fungicides resistance appearance of Fusarium sp. ?

L61 : change the term “pathways” to “mechanisms”

Thank you for taking the time to reread this article, there are some small errors: metabilite (l493)....

Author Response

Reviewer 3 Report (Previous Reviewer 3)

Please mention which endogenous control was used for qPCR and what method was used for interpreting the results (ddCt?). 

This is the second round of revision and I see that the authors did all required comments. I still see that qPCR requires more details in the methods section. 

Author Response

This manuscript is a resubmission of an earlier submission. The following is a list of the peer review reports and author responses from that submission.

Round 1

Reviewer 1 Report

The article describes some mechanisms that could be involved in the growth inhibition of Fusarium asiaticum and the DON formation in grains. Overall, the methodology used by the authors is correct and the data is robust. However, there is a need to improve the explanation of certain results, mainly concerning the description of the images presented.

Below I detail some recommendations to improve the article:

Line 66: The authors consider a fourth objective "To use the real-time fluorescence quantitative polymerase chain reaction (RT-qPCR) method to prove the accuracy of the obtained transcriptomic data." I think that it should not be included since it is part of the validation of objective N°3.

Line 103-117. In the paragraph, the authors propose two techniques to study the Effect of B. velezensis E2 on F. asiaticum F1 Growth: plate confrontation and inhibition of spore germination. However, the wording is confusing and the paragraph must be rewritten. Is the plate confrontation carried out using F. asiaticum plugs, disks soaked in a conidia suspension, or 30 ul in each well? How many repetitions of the plate confrontation were performed?

Line 182: Is the subtitle "Primer design" correct? I think the subtitle does not match the paragraph described between lines 183 and 188.

Line 203-207. Figure 1. The description and quality of the figure are fragile and should be improved. Figure 1a) What do the authors mean by "initial form"? How many days of incubation do the cultures in the image have? What culture medium is used? Fig 1b) What magnification do you use? Fig 1d) The spikes shown correspond to how many days of post-inoculation of the pathogen? What symptoms are observed? Highlight important areas in the images using arrows or boxes to draw the reader's attention and explain what they are looking at.

Line 216: The description of the figure should be improved.

Line 225: Figure 3. The description in the figure caption should be improved. Fig 3a) The description in the figure caption should provide the following information: what culture medium was used? What does each well correspond to? How many days of incubation does the plate shown in the image have? It is not clear whether the controls are included in the same PDA Petri dish. Fig 3b) Indicate using arrows the areas to highlight in each image, such as the vesicle-like structures mentioned in the text.

Line 233. Highlight the areas with arrows or boxes in each image to focus the reader's attention. It is not clear what should be observed in each figure. It is necessary to indicate the areas to highlight.

Line 330. Figure 8. To improve the quality and image resolution. Increase the font size of the text in the figure axes and legend to make it easier to read.

Line 349. Use italics for Bacillus amyloliquefaciens.

Line 355: Capitalize "The GO".

Reviewer 2 Report

Manuscript jof- 2934542 : Control of Fusarium head blight of wheat with Bacillus velezensis E2 and potential mechanisms of action.

The work carried out in this study is very interesting: firstly, in view of the model used and the problem represented by Fusarium Head Blight in wheat grain production and, secondly, in view of the data provided by the transcriptomic study carried out on Fusarium asiaticum in response to Bacillus velezensis, a potential biocontrol agent. Nevertheless, the writing is not sufficiently careful or informative about the methods and in the accuracy of the results obtained.  I cannot therefore accept it as it stands. I hope that my comments will help to improve this article.

Abstract :

L12 : change “Wheat plants are easily inhibited by...” to “Wheat plants are impacted by…”

L13-L14 : Move and change the sentence “ in this study, a fungal”.

“…. development, storage and food safety. In this study, a fungal strain was isolated from diseased wheats and identified as Fusarium asiaticum F1, known to be a member of the Fusarium graminearum species complex, agents causal responsible of FHB. In order to control this disease, new alternatives need to be developed as the use of antagonistic bacteria. Bacillus velezensis…”

L17 : replace “(DON) formation in grains.” by “(DON) synthesis in grains.”

Maybe add elemeents on the problematic of toxns and the risks of these ones on humans (just one sentence !)

L18 : add a “space” “F1was” to “F1 was”

L25 : replace “inhibit  the gene expression” by “reduce the gene expression”

Introduction : Please add a few points to your introduction

In paragraph 31-36 : Please add information on crop (percentage) or economic losses due to disease. If it’s possible add the names of the various fungi implicated in the FSG complex.

L37 : change the sentence to : “Fusarium mycotoxins are a group of toxic secondary metabolism secreted by Fusarium sp., such as DON …….”.

Paragraph 37-45 : what does "largest pollution rale 1 : Add a more complete title, and give the reference of the article from which the primer sequences were extracted. nge" mean ? Thank you for using the term contamination rather than pollution. Reformulate the sentence.

L44-45 : thank to add some percentage of actual grain contamination...

L47-48 : You seem to be saying that biological control of Fusarium is as widely used as chemical or breeding plant... Is this the case? What is the novelty of your work if biological control already exists? Maybe reformulate correctly the problematic.

Paragraph 47-56 : No trials with B velezensis or other Bacillus sp. against fungal diseases in wheat ? no trials against Blumeria graminis f.sp tritici (mildew) or Septoria (STB) in the litterature ?

Material and Methods Part :

L83-84 : Add the sequence of the primers : ITS and TEF-α

L96 : Title of Table 1 : Add a more complete title, and give the reference of the article from which the primer sequences were extracted.

In the "toxigenic chemotype" column, add the full name of the toxins sought and their abbreviation

Paragraph 92-95 : please add a sentence explaining that the PCR products are then plated on a gel (agar percentage) to identify the presence of the various amplicons sought.

L100-102 : Thank you for detailing the method of cultivation (medium, time, concentration…) and the concentrations of bacteria used, these are important elements to give to the readers by extracting them from your referenced article !

L104 : Thank you for detailing the plate technology used, there aren't enough details to make it easy to understand the method used, 30µl per well (concentration of bacteria??).

L111 : 100µl of antagonistic antibiotic ? what is it ? 100µl of antagonistic bacteria ? concentration ?

L113 : about 200 pathogens ? I don't understand, are you talking about testing more than 200 strains of F. asiaticum? or 200 observations of spores? 200 spores?

Results :

L96 : Thanks to use a more fuller and informative title

Figure 2 : Complete the figure by explaining in the legend or on the gel what is in each column (there are three columns + the size marker column... what are these three columns?). Add an arrow for the expected amplicon sizes: 234 and 415, 586 and 864.

Paragraphe 131 – 142 : Thanks to reformulate some sentences, it’s not clear. Please avoid using parentheses to explain the dilution, some sentences have no verbs…

L133 : Thank to reformulate the sentence : “5 g of decluttered was wheated and placed in 150 mL”

L135 : B. velezensis and not B. Velezensis…

Paragraph 144-157 : I'm sorry, but I can't understand when Fa is exposed to the bacteria, for how long and at what concentration for each organism. There is no mention of how the RNA is extracted and sequenced ... changed the title, which has absolutely nothing to do with this part... and where is the extraction of this RNA mentioned in this article? Protocol ?

L148 : why “then” ?

L159 : change “detected” to “verified”.

L178 : You quoted a reference for the design of the primers, but if these are new primers specifically made for this work, you must clearly give the software used and the subsequent checks on their specificity.

Table 2 : please add their full names followed by their abbreviations to the column of gene names.

L175 : change the title !

L182 : change the title ! This title would be more appropriate for the paragraph just before it…

L183 : F. asiaticum in italic

Paragraph 183-187 : Please add the references of the instruments used and at least the names of the different kits used, if not the complete protocols... 

Results :

L193 : change the title to something more informative

L194 : and the ITS ?

L203 : the initial form ? What does this mean, that the strain has evolved (mutation?) the duration of your manipulations?

L209 : F1 or the ADN of the fungi ? It’s not clear, thanks to be more specific !

L221 : change “the number of germinations”  You talk about numbers, but have you assessed them on the basis of a precise number of events : number of germinated spores versus number of non-germinated spores? If so, what are the rates or data ? It would be good to have germination rates as well as photos...

L235 : change the title, you're not talking about growth, just the production of fungal toxins (Table 3 and not the growth)… maybe it’s not the good table added ?

L246 : I’m not satisfied for the control used, for me the exact control was F. asiaticum with the same quantity of medium culture bacterial-free added (Luria-Bertani medium)….

Part on the analysis of DEGs : I'm sorry, but I'm a bit lost in the figures in this section.... Authors explained that 3482 genes are differentially expressed with 2071 down and 1411 up. L262 : ayhirs explained that among these, 3414 are annotated (GO terms) : 2817 in biological process, 1181 in molecular function, 2144 cell components category…. Then 2817+1181+2144 = 6142 and not 3414 ? Thanks to be more precise, maybe same DEGs belong to more than one category?

Paragraph 304-308 : no mention about the expression of the genes up-regulated ?

L314 : “and the pathogenic bacteria?” but you work on a pathogenic fungi… I’m not well understand the mention…

L320 : change “F.asiaticum” to “F. asiaticum”

L334 : please provide some data on quality or yield losses

L338 : replace “pollutes” to “contaminates”

Paragraph 338-340 : it's too generalist (other plants) thank you for sticking to wheat and giving more details! Different types of toxins and at what rate... etc

L343 : Many studies have found that Bacillus has an obvious in inhibitory effect on Fusarium… and so what is the contribution of this work? Thank you for formulating a sentence at the beginning of the discussion showing the novelty of your work to what is already known...

Reviewer 3 Report

First, I would like to thank the authors for the transcriptome study they provide in this manuscript. The manuscript entitled “Control of Fusarium Head Blight of Wheat with

Bacillus velezensis E2 and Potential Mechanisms of Action” studies the antagonistic potential of Bacillus velezensis E2 isolate against Fusarium head blight disease in wheat. The study included biological profiling of the direct microbial interaction and the nature of action that the bioagent do against the fungal pathogen. Additionally, an RNA-seq analysis was done to fortify the topic and pinpoint the mechanistic pathways that are related to this antagonistic interaction. This study showed the involvement of four pathways in response to the antagonistic interaction.

However, I have few comments on the hypothesis, methods, and the conclusions.

1-       INTRODUCTION presents sufficient information on the topic and covers all aspects of the study, but I would like to see a brief about the genes/pathways in the fungal pathogen that are triggered upon antagonistic interaction and what is their role..

2-       MATERIALS & METHODS: I would like to see more detailed methods describing the following:

a.        sampling procedures, (Line 72),

b.        the observed symptoms on collected samples,

c.        what media was used for isolation (Line 80),

d.        either primer sequences or references (Line 84),

e.        steps of pathogenicity test (Line 90).

f.           The method of antagonistic interaction in line 102.

g.         Please write potato dextrose agar (PDA) Line 104

h.        What is the source of antibodies in Line 140

i.            Please write the full name of RT-qPCR Line 183

j.            Please mention the internal control and describe the method used for normalization.

k.         What statistical method did you use first? (t-test?)

3-       RESULTS:

a.        Please make titles more meaningful

b.        Fig.1: please define the outgroup with its accession number and specify the bootstrap value.

c.        Fig 2: please give more details on the expected amplicon size, the ladder used, and agarose percentage.

d.        Fig. 3 & 4: please add arrows to the fungal hyphae that were affected upon antagonistic interaction. Please write what is SEM Line 233

e.        Table 3: please describe what is control and what is treatment groups.

f.           Fig 5: please specify what software generated the volcano plot, and briefly describe what are the inputs.

g.         Fig 6, 7 & 8: please briefly describe what are the inputs.

4-       Discussion: requires support by more references. Please discuss more about the roles of the discovered pathways because of antagonistic reaction in fungi, and relate it to the aim of the study.

Please look at my comments in the PDF file.
